# AN INVESTIGATION INTO THE ROLE OF AUTHOR DEMOGRAPHICS IN ICLR PARTICIPATION AND REVIEW

## ABSTRACT

As machine learning conferences grow rapidly, many are concerned that individuals will be left behind on the basis of traits such as gender and geography. We leverage historic ICLR submissions from 2017 to 2021 to investigate the impact of gender and country of origin both on representation and paper review outcomes at ICLR. We also study various hypotheses that could explain gender representation disparities at ICLR, with a focus on factors that impact the likelihood of an author returning to the conference in consecutive years. Finally, we probe the interplay between paper topic and review outcomes, and perform a study on how the inclusion of theorems in a paper and the size of the author list correlates with paper scores.

## 1 INTRODUCTION

It is well known that the field of computer science has strong representation disparities across both racial/ethnic and gender categories (Zweben & Bizot, 2021; Muro et al., 2018; Dillon Jr et al., 2015). For example, women comprised just 23.6% of enrollees in computation graduate programs in 2020, and only 1.2% of awarded PhDs were to Black students (Zweben & Bizot, 2021). While these statistics paint a concerning picture of the field's ability to recruit and retain students from diverse backgrounds, representation at ICLR is considerably more distorted. Last year, Tran et al. (2020) reported that only 12.1% of ICLR papers from US universities were lead by a woman first author. Using methods described in this paper, we observe that this number decreased to 10.62% in 2021.

Gender representation disparity is quite extreme at ICLR, even within the context of well-known disparities in computer science at large. This motivates us to study these representation disparities more deeply, and we seek to identify and analyze theories for why they might exist. In particular, we are interested in whether issues concerning authorship, community, and the review process can be identified as factors that influence the representation of different groups. We consider these issues:

- **Retention** We find that women who attend ICLR are less likely to return to the conference the following year than men. We study a number of factors that may contribute to this effect. We find that women are much more likely to return when they are mentored by a woman and that both women and men are far less likely to return after receiving negative scores during review. At the same time, women are advantaged by working with slightly larger authorship teams, which tends to result in more positive outcomes.

- **Country of Origin** We find that papers from western nations tend to score higher, while papers from East and South Asia score lower than the conference average, although no significant bias against Asian papers is detected when acceptance decisions are made, controlling for reviewer scores.

- **Topic Breakdown & Theory vs Empirical papers** We see that papers containing theorems are far more likely to be accepted than non-theory papers. Further, women tend to submit papers to topics with slightly lower acceptance rates than men.

- **Industry Affiliations** Papers from Google, Facebook, and Microsoft are much more likely to be accepted on average. Interestingly, women first authors are slightly more prevalent among industry papers than among academic papers.

While the focus of our study is on demographic differences in review outcomes, a number of recent studies have been conducted on the review process at large, with a focus on reproducibility and quality of reviews (Rogers & Augenstein, 2020; Bharadhwaj et al., 2020; Stelmakh et al., 2020b;a; Fiez et al., 2020; Manzoor & Shah, 2020; Shah et al., 2018; Stelmakh et al., 2019).

## 2 DATASET CONSTRUCTION

Our analysis is based on a dataset we acquired using the APIs of *OpenReview* and *SemanticScholar*, followed by extensive hand-labeling procedures conducted by the authors. From *OpenReview*, we obtained titles, paper text, author lists, emails, scores, and reviews for ICLR papers from 2017-2021. In total, 8,553 submissions were obtained: 2,978 from ICLR 2021, 2,560 from ICLR 2020, 1,565 from ICLR 2019, 960 from ICLR 2018, and 490 from ICLR 2017. To identify an author's institution, we used the author's email listed on their *OpenReview* profile and then mapped them to institutions with the open source World University and Domains dataset.

We used *SemanticScholar* to scrape author- and paper-specific data. Author-specific data includes the number of citations, publications, influential papers, and h-index of each author on the paper. These data was obtained by scraping every paper by each author on *SemanticScholar*, identifying papers with titles that were either identical or similar as measured by edit distance, and hand-validating to see if identified papers were correct. Paper-specific data includes the number of citations.

To study the role that gender plays in the review process, we hand-labeled gender for first and last authors on ICLR 2021 submissions. These labels were produced without relying on an automated process and rather by searching for gendered pronouns that appeared on personal webpages, social media, and CVs if applicable, and choosing the canonical gender for the name otherwise. We chose this process because ICLR has broad international participation, and automated gender inference tools, while more reproducible, are known to have unusually strong biases and inaccuracies on non-western identities (Santamaría & Mihaljević, 2018). We consider only binary gender labels. Future work should consider the experiences of non-binary and transgender people, however these populations cannot be easily studied using the statistical methods used in this paper.

## 3 GENDER REPRESENTATION AT ICLR

Tran et al. (2020) identified that women are highly under-represented at ICLR; in 2019, women made up 23.2% of all CS PhD students in the US (Zweben & Bizot, 2020), while only 12.1% of first authors at ICLR from US universities were women. We find that this disparity widened in 2020, with women compromising just 10.62% of first authors and 10.77% of last authors. For comparison, women comprised 23.4% of enrollees in doctoral CS programs and 24.8% of computer science, computer engineering, and information programs combined (Zweben & Bizot, 2021).

There are many complex social and economic factors that likely contribute to the under-representation of women in computer science, including confidence gaps between men and women in STEM classrooms (Lubienski et al., 2013), stereotypes of skill levels and gender roles in academics (Lubienski et al., 2013), differences in workplace treatment between men and women (Funk & Parker, 2018), and even differences in income after graduation (Xu, 2015). However, the representation disparity at ICLR is strikingly more lopsided than what is observed in the computer science community at large, which indicates community-specific factors that influence representation.

In this section, we consider various hypotheses for why such a strong disparity in representation might exist at ICLR. While there may be a range of sociological factors at play that are outside the scope of our study, we focus on factors that can be analyzed through publicly available data. Throughout this section, we focus on the likelihood of women returning to ICLR after appearing once. We focus only on the return rate of first authors. This is because last authors are unlikely to be first time attendees and often have long professional careers in ML that make them likely to return to the conference regardless of the factors identified below.

### 3.1 RETENTION

We observe that ICLR has somewhat more difficulty retaining women participants than men. Among first-time conference attendees in 2020, of the women who appeared as first author on a paper, 19.79% returned in 2021, compared to 23.83% for men. Expanding to account for all 2020 attendees regardless of their number of appearances, this mild discrepancy remains; the return rate for women was 28.12% compared to 31.64% for men.

As for why this discrepancy exists, the difference in return rate could be explained in part by outcomes in the review process. We hypothesize that authors receiving strongly negative feedback on their contributions may be less likely to return to the conference and that the relatively lower scores women receive make them less likely to return. Tran et al. (2020) reported that papers from women first authors on average scored 0.16 points lower than men in 2020. Our 2021 data shows a present but smaller gender gap with women first authors on average scoring 0.13 points lower than men first authors ($p = 0.03$). This gender discrepancy is clearly visible in the score histograms in Figure 1. There is no discernible gender gap for last authors.

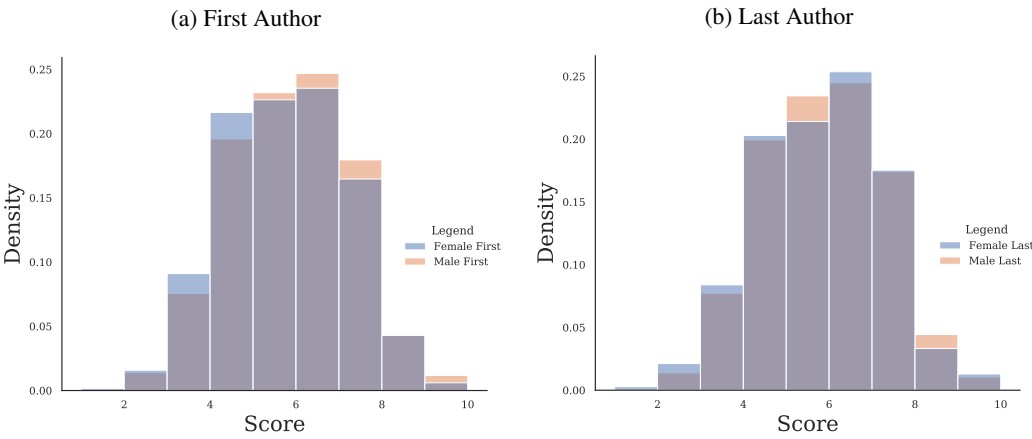

Figure 1: Histogram of average scores by gender, first and last author, ICLR 2017 - ICLR 2021. We observe a pronounced gap in first author scores, but a mostly equal distribution for last authors.

This hypothesis is indeed supported by the data – review scores strongly correlate with retention rates. A logistic model predicting author return probability as a function of mean reviewer score and gender identifies reviewer scores as a strong factor in predicting author return ($p = 0.015$, Table 1). This trend is a significant factor within both the men and women sub-populations (see Table 5 in the Appendix). In a separate test, women with rejected papers had a return rate of 24.28% while men with rejected papers had a return rate of 30.28% ($p = 0.073$).

Research on small minority peer groups in STEM fields (Etzkowitz et al., 1992; Rosser & Lane, 2002) suggests that the effect of reviewer scores may be stronger for women than for men, however we did not detect this as a significant factor in Table 1).

Table 1: **Return rate for first authors as a function of mean reviewer score and gender indicator.** The logistic regression model had a 70.8% accuracy and a hold-out set containing 30% of 2,122 first authors.

| Variable | Coefficient | Std. Error | Z-score | p-value |
|---|---|---|---|---|
| mean reviewer score | 0.236 | 0.097 | 2.430 | 0.015 |
| gender indicator | 0.374 | 0.460 | 0.811 | 0.417 |
| gender × mean review | -0.045 | 0.102 | -0.441 | 0.659 |
| constant | -1.960 | 0.437 | -4.488 | 0.000 |

## 3.2 Amount of professional experience

Negative reviewer scores are commonplace in the machine learning community at large. One may expect experienced authors to be well acquainted with this fact, and therefore less strongly impacted by negative scores than those authors receiving their first ever round of reviews. In this section, we quantify how author experience correlates with the return rate of first authors. We consider the relation between professional experience and first author outcomes, in addition to last author's experience level.

We identify "first-time" authors as people who did not submit a paper to ICLR in 2017-2019. Note that we use "first author" to refer to author order, and "first time author" to refer to lack of prior submissions. We find that first time authors are less likely to return to ICLR than other authors ($p<0.001$). This effect is still highly significant even after controlling for reviewer scores. Women first-time authors return with a rate of 20.36%, while men first-time authors return with a rate of 24.54%. The observations in this study are applicable to many attendees, as 65% of women and 63% of men are first-time authors.

We also consider the number of publications an author has and how it correlates with their return rate. After controlling for reviewer scores[1], we find that the number of publications of the first author was correlated with return rate, but not to a statistically significant degree ($p = 0.143$, Table 6). We find that the number of citations is not a good indicator of return rate, possibly because it varies wildly for young researchers.

## 3.3 Mentorship and Gender pairing

Gender researchers in STEM fields have often cited the gender pairing of mentors and mentees as a major factor in predicting author productivity (Gaule & Piacentini, 2018; Pezzoni et al., 2016; Kato & Song, 2018). At ICLR, we find that first authors are more likely to return to the conference if their mentor is of the same gender as they are. Women first authors with women last authors had a return rate of 42.22% while women first authors with male last authors had a return rate of only 25.40% ($p = 0.02$). A similar trend existed for men, who returned 46.36% of the time with a male last author, and only 31.01% of the time with a women last author ($p<0.001$).

Curiously, women-women teams had a lower acceptance rate (23.5% vs 27.3%) than women-man teams, and yet women-women teams still result in much higher retention. Furthermore, when we control for paper scores, a logistic model still identifies gender pairing as yielding a highly significant boost to return rate (p=0.01).

Finally, we see that the experience of the last author, as measured by the log of their total citation count, correlates strongly with first author return rate. After controlling for reviewer scores, we find that the citation count of the last author is a significant factor in predicting first author return rate, with more experienced last authors making return more likely ($p = 0.002$; Table 7).

## 4 Authorship teams

In this section, we investigate whether larger authorship teams perform better. Our study is motivated by the prevalence of personal accounts of isolation among women in computer science (Sankar, 2015; Fisher & Margolis, 2002; Frenkel, 1990) which posits that the relatively smaller peer group for women may result in having fewer collaborators, and as a result fewer co-authors. Our study does not find evidence of this and, in fact, indicates that women tend to have larger authorship teams than men.

For this study, we consider papers with 1 author, 2 authors, 3 authors, 4 authors, 5 authors, 6 authors, 7 authors, and 8 or more authors. These categories had 274, 1396, 2144, 1896, 1276, 767, 410, and 383 submissions, respectively.

We find that mean reviewer scores increase monotonically with the number of authors (Figure 3). This is consistent with published observations that collaborative papers in the sciences generally

---

[1]Less experienced authors are likely to receive lower scores, making them less likely to return. Our model includes reviewer score as a control variable to remove this effect.

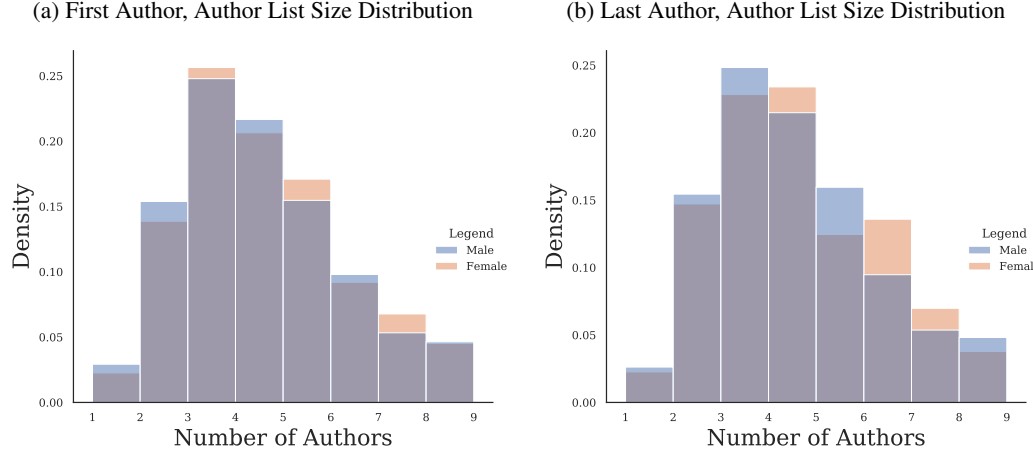

(a) First Author, Author List Size Distribution    (b) Last Author, Author List Size Distribution

Figure 2: Distributions of number of co-authors based on gender. With respect to the gender of (left) first author (right) last author.

score better and accrue citations faster than single author papers (Shneiderman, 2018; 2016; Perkmann et al., 2013). A regression model controlling for reviewer scores does not find a significant correlation between author number and acceptance rate, i.e., ACs do not appear to preferentially accept papers with more authors.

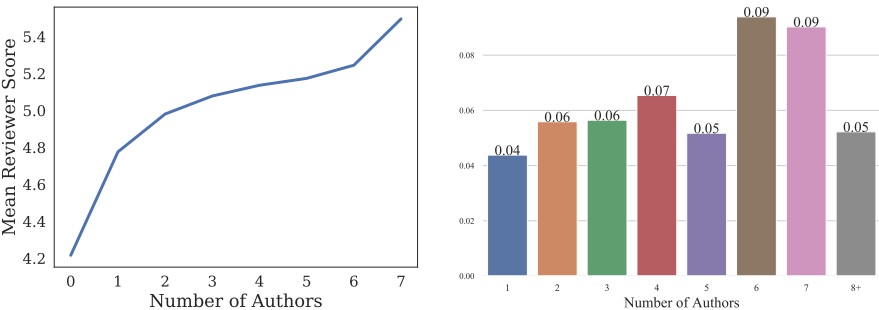

Figure 3: (left) Mean reviewer scores as a function of number of authors, ICLR 2017 to ICLR 2021. (right) Percent of women-led papers in each authorship category, ICLR 2017 to ICLR 2021.

It might be possible that papers with more authors are more likely to be accepted due to the higher probability of personal connections with an AC. We do not find enough support for this hypothesis; a regression model controlling for reviewer scores does not find a significant correlation between author number and acceptance rate, i.e., ACs do not appear to preferentially accept papers with more authors.

We find that women on average were part of larger authorship teams. Men first author papers had on average 4.06 authors, while women had 4.25. Men last author papers had 4.06 authors on average, Women had 4.37. See Figure 2 for the distributions.

## 5    GEOGRAPHIC REPRESENTATION

This section breaks down how a paper's country of origin correlates with review outcomes. We focus on academic (rather than industrial) papers, as they are associated with a university with a well-defined location. We attribute a paper to the national affiliation of its last author. We separate the papers into ten different regions: Australia and New Zealand (96 papers), Canada (333 papers), East Asia (including China, Japan, Mongolia, North Korea, South Korea, and Taiwan, 498 papers),

Mainland Europe (498 papers), the Middle East (including Bahrain, Cyprus, Egypt, Iran, Iraq, Israel, Jordan, Kuwait, Lebanon, Oman, Palestine, Qatar, Saudi Arabia, the Syrian Arab Republic, Turkey, the United Arab Emirates, and Yemen, 113 papers), Russia (13 papers), South America (13 papers), South Asia (including Afghanistan, Bangladesh, Bhutan, India, Nepal, Pakistan, Sri Lanka and the Maldives, 355 papers), the UK and Ireland (438 papers), and the United States (2345 papers).

Charts showing the geographic distribution of submitted and accepted papers appear in Figure 4. Papers from the US and Canada on average scored 0.1 and 0.22 points higher than the conference average respectively, while papers from East Asia and South Asia scored 0.36 and 0.55 points lower than the conference average, respectively. Other regions did not deviate from the average score in a substantial way. The acceptance rate of the conference over the years studied was 30.19%, while the acceptance rate for the US was 34.58% ($p < 0.001$), Canada was 35.73% ($p = 0.012$), East Asian was 19.27% ($p < 0.001$), and South Asia was 22.53% ($p < 0.001$). See Table 10 in the appendix.

A logistic model that predicts acceptance as a function of mean reviewer score and an indicator variable for East and South Asia indicates that, controlling for scores, papers from East and South Asia were slightly less likely to be accepted by ACs. This discrepancy in acceptance rates is equivalent to a change in mean reviewer score of 0.098 and had borderline p-value 0.11 (Table 2).

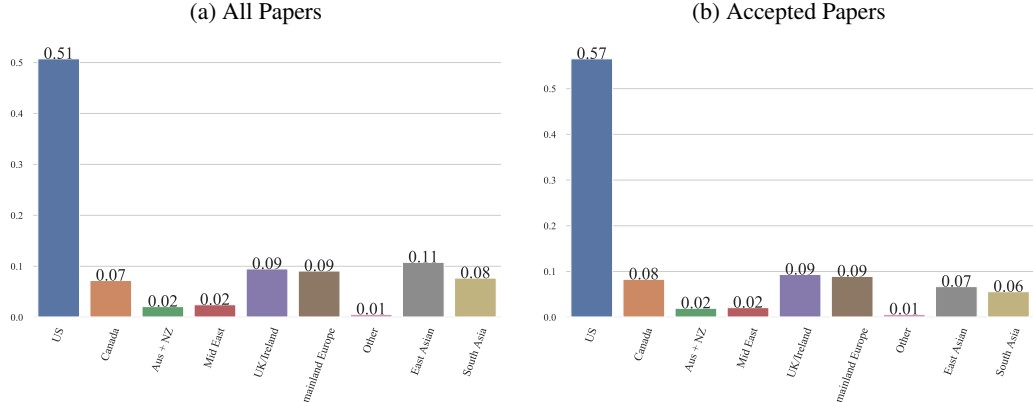

Figure 4: Geographic distribution of papers submitted and accepted

It appears that the scoring difference between papers of Western and Asian origins is likely explained in part by Asian authors having less professional experience on average than authors from other regions. 84% of papers of East and South Asian origin were lead by a first-time author, whereas first-time authors lead 66% of papers from the US. We also suspect that scores are affected by gaps in English language experience in addition to cultural differences in how non-Western speakers write and communicate, although we are not able to quantify these latter impacts.

Table 2: **Impact of East Asian origin on acceptance**, all submissions to ICLR from 2017 to 2021. Logistic regression predicting paper acceptance as a function of mean reviewer score and an indicator of whether the paper was from East Asia. The model achieves 90% accuracy on a hold-out set containing 30% of 2021 papers.

| Variable | Coefficient | Std. Error | Z-score | p-value |
|---|---|---|---|---|
| mean reviewer score | 3.0912 | 0.074 | 41.506 | 0.000 |
| east asia indicator | -0.2880 | 0.180 | -1.598 | 0.110 |
| constant | -18.2774 | 0.437 | -41.796 | 0.000 |

## 6 TOPIC DISTRIBUTION

To understand how the topic of a paper correlates with the likelihood of it being accepted, we define a high-level categorization of papers based on hand-curated keywords. When any of these keywords appear in the paper title and/or abstract, we assign the topic to the

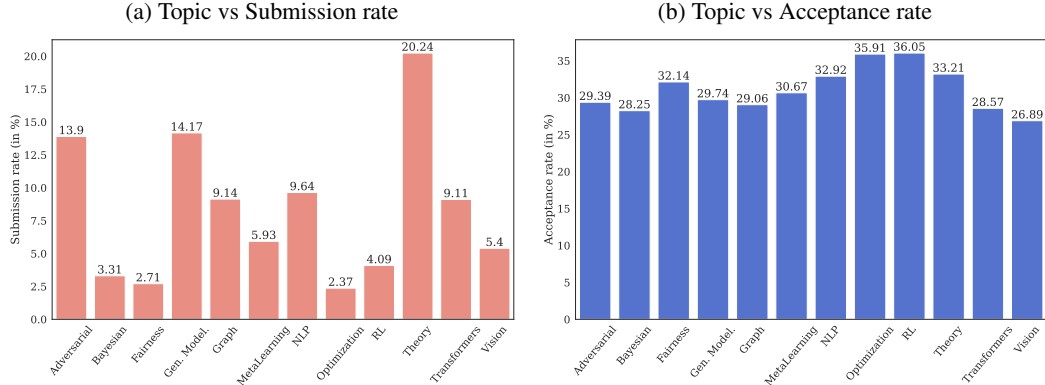

Figure 5: Topic level submission and acceptance rates on papers submitted from ICLR 2017 - 2021.

paper (some papers appear in multiple topics). The high-level topics used in this study are: Adversarial, Bayesian, Fairness, Generative models, Graph ML, Meta-Learning, NLP, Optimization, Reinforcement Learning (RL), Theory, Transformers, and Vision. We refer the reader to Appendix A.1 for the keywords used in this study.

In Figure 5a, we show the percentage of papers falling into each of the topic categories. The numbers are computed on ICLR papers from 2017 to 2021. Theory forms the biggest chunk followed by Generative models and Adversarial ML. In Figure 5b, we show the percentage of accepted papers for a given category. Reinforcement Learning was the most successful topic followed by Optimization and Theory. We see in Fig. 6a that women were less likely than men to submit a paper in these three categories and more likely to submit in Vision and Transformers which were the least successful topics at ICLR. When it comes to topic level acceptance rates, women did better than men in Meta-Learning, Fairness, and Theory and almost comparable in Vision (Fig. 6b).

We also examine how paper submission rates per topic changed based on the geography of the first author in Figure 8 (left). We examine 4 regions - USA, Canada, Europe, Asia. US submitted the highest proportion of papers in Theory, while Asia was highest in NLP, Transformers, Europe in Bayesian, Generative Models and Canada in Theory, Meta-Learning, and Fairness.

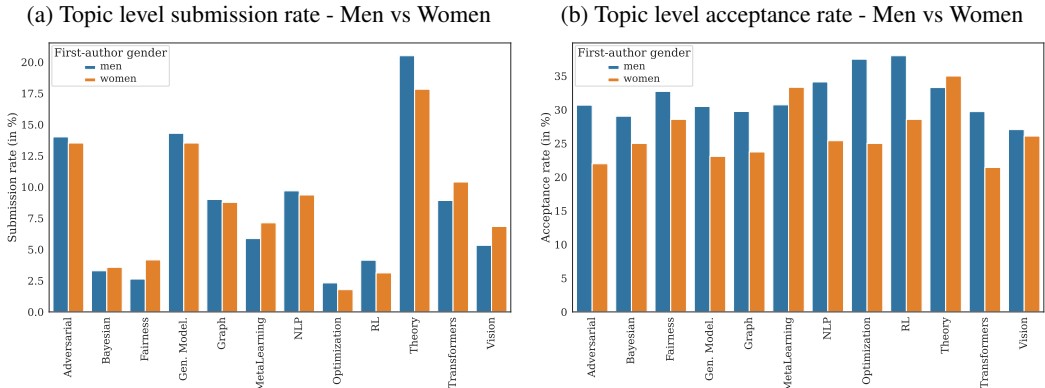

Figure 6: Topic breakdown by gender, submission vs accepted papers, ICLR 2017 - ICLR 2021.

## 6.1 DO PAPERS WITH THEOREMS DO BETTER THAN EMPIRICAL PAPERS?

It is common across sub-fields to categorize papers as theory papers and empirical papers, and it is colloquially thought by many that papers containing a theorem are more likely to be ac-

cepted to a conference than purely empirical studies. We classify a paper as a theorem paper if the paper body contains any of the following patterns: `"Theorem:"`, `"Theorem <number>"`, `"Proposition:"`, `"Proposition <number>"`, `"Lemma:"`, `"Lemma <number>"`, Out of the 8553 papers in the data set, 2321 submissions had at least one of the aforementioned patterns.

Theorem papers on average scored 0.34 points higher than papers classified as non-theorem papers (p<0.001). The score distribution for the two kinds of papers is depicted in Figure 7. In all paper topics, papers with theorems out-performed those without theorems, although not always by a statistically significant margin, with Fairness and NLP getting the biggest boosts (Table 12). Among women first authors, 26.7% of papers contained a theorem compared to 31.1% for men ($p = 0.028$). Finally, although theorem papers may receive higher mean reviewer scores, they do not receive a bonus at the AC level (Table 3). We also examine how likely researchers from different demographics added a theorem in papers from different topics in Figures 7 (right) and 8 (right). We see women were more likely to have a theorem in `Theory` and `Fairness` topics, while men were more likely to have one in `Optimization` and `RL` topics. When it comes to region-level trends, the US was more likely to have a theorem in most of the topics while Canada was less likely.

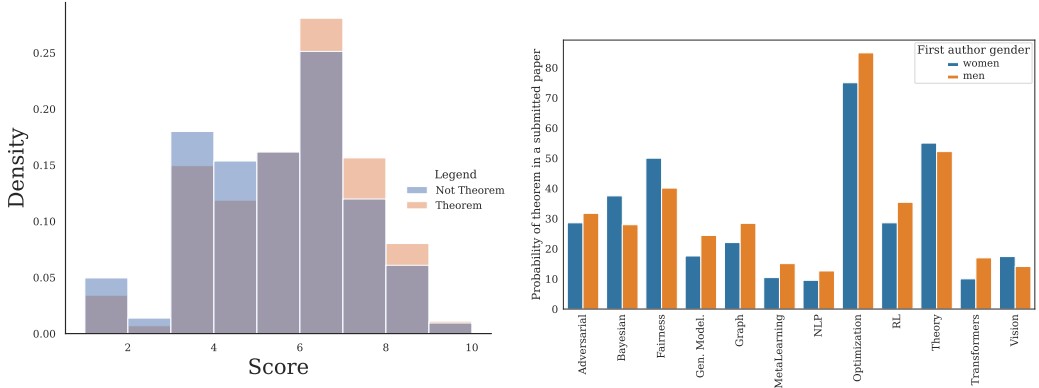

Figure 7: (left) The distribution of paper scores with and without a theorem. (right) Fraction of theorem papers by topic - Men vs Women

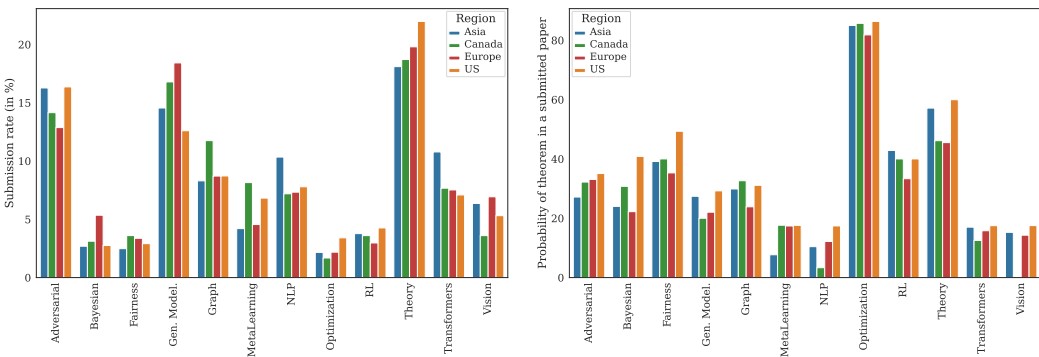

Figure 8: In this figure we study topic distribution with respect to geography. (left) Fraction of papers by topic across regions (right) Fraction of theorem papers by topic across regions.

## 7 HOW DO INDUSTRY PAPERS REVIEW?

In this section, we break down the review statistics for industry papers submitted and published by Google (including DeepMind), Facebook, and Microsoft. In 2021, we find that papers from each of these organizations scored better on average than a typical conference paper. Without controlling for scores, all three organizations were significantly more likely the have papers accepted in 2021. The conference acceptance rate for 2021 was 28.87%, while the rate for Google was 43.29% (p<.001),

Table 3: **Theorem impact on acceptance**, all submissions to ICLR from 2017 to 2021. Logistic regression predicting paper acceptance as a function of mean reviewer score and a theorem indicator. The model achieves 89% accuracy on a hold-out set containing 30% of 2021 papers.

| Variable | Coefficient | Std. Error | Z-score | p-value |
|---|---|---|---|---|
| mean reviewer score | 3.0926 | 0.075 | 41.470 | 0.000 |
| theorem indicator | 0.0518 | 0.085 | 0.609 | 0.542 |
| constant | -18.3113 | 0.438 | -41.795 | 0.000 |

Table 4: **Industry impact on acceptance, Google, Facebook, and Microsoft**, ICLR 2021. Logistic regression predicting paper acceptance as a function of 3 industry indicators. The model achieves 70% accuracy on a hold-out set containing 30% of 2021 papers.

| Variable | Coefficient | Std. Error | Z-score | p-value |
|---|---|---|---|---|
| google | 0.7050 | 0.110 | 6.428 | 0.000 |
| facebook | 0.4859 | 0.168 | 2.896 | 0.004 |
| microsoft | 0.3573 | 0.166 | 2.147 | 0.032 |
| constant | -25.6433 | 0.636 | -40.350 | .000 |

Facebook was 38.80% ($p = 0.004$), and Microsoft was 42.42% ($p = 0.032$). Women made up slightly more first authors on industry papers (12.86%) than they did on all papers (10.62%).

Tran et al. (2020) finds that, controlling for reviewer scores, ACs were relatively less likely to accept a paper from Microsoft (-.50 points, $p = 0.003$) while submissions from Google (.007 points, $p = 0.932$) and Facebook (.07 points, $p = 0.630$) received non-significant acceptance boosts at the AC level at ICLR 2020. When controlling for score using the same procedures as Tran et al. (2020) for consistency, we find that at ICLR 2021, although the negative bias towards Microsoft submissions disappeared (.08 points, $p = 0.204$), a positive bonus on Google submissions surfaced (.118 points, $p = 0.008$). See Table 13.

## 8 CONCLUSION

Using publicly available historical data and hand-curated datasets, this article examines the interplay between demographics, representation, and review outcomes at ICLR. This study has several important limitations that must be acknowledged. First, this is an observational study where we were not able to do controlled experiments or identify causal relationships. Second, non-i.i.d. review scores caused by people reviewing multiple papers, and violations of parametric assumptions, can cause Type-I errors. These issues have been particularly problematic when detecting bias in reviews (Stelmakh et al., 2019). Finally, note that ICLR was a virtual conference in 2020 and 2021, which may impact demographics. Furthermore, while we did find significant results for some factors, we also report trends that, in some situations, did not rise to the level of statistical significance. Unfortunately, our sample size is limited by the number of submissions, and the relatively small number of women at ICLR limits the statistical power of our tests.

The study has two major goals: to raise awareness of representational issues at ICLR and to produce actionable conclusions. One of the strongest actionable conclusions reached in this paper is that *mentorship is important*; women were far more likely to return to the conference when working with a women last author (25% vs 42%). Furthermore, we suspect much of the disparity in acceptance rate of papers from Asia is due to English language experience – another factor that can be addressed by mentorship. Finally, we know that first authors paired with more experienced last authors scored better (which promotes retention) and have improved retention rates overall. Collectively, this evidence supports the use of mentorship and role model pairing programs to enhancing representation.

## 9 ETHICS STATEMENT

This study was conducted using publicly available datasets and without surveying participants. For this reason, we do not consider this study to raise privacy or data security issues, or to constitute human subject research in the US federally defined sense.

Finally, there is a risk that our observations of lower reviewer scores among some sub-populations may promote stereotypes or reinforce negative opinions. We emphasize here that correlation does not imply causation, and the negative effects observed in this study are the result of confounding variables, some of which we tried and succeeded to identify, and some of which we did not.

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

## A APPENDIX

### A.1 BREAKDOWN OF PAPERS BY TOPIC

The topics [and keywords] used were `Vision` [computer vision, object detection, segmentation, pose estimation, optical character recognition, structure from motion, facial recognition, face recognition], `NLP` [nlp, named-entity, machine translation, language model, word embeddings, part-of-speech, natural language, BERT, GPT], `Meta-learning` [few-shot, meta learning, transfer learning, zero-shot], `Adversarial ML` [adversarial attack, poisoning, backdoor, adversarial example, adversarially robust, adversarial training, certified robust, certifiably robust], `Generative modelling` [generative adversarial network, gan, vae, variational autoencoder, diffusion models], `Fairness` [gender, racial, racist, biased, unfair, demographic, ethnic, ethical], `Optimization` [optimization theory, convergence rate, convex optimization, rate of convergence, global convergence, local convergence, stationary point], `Graph` [graph, tabular], `Transformers` [attention, transformer], `Bayesian` [bayesian], `Reinforcement Learning (RL)` [temporal difference learning, value function, value network, policy gradient, actor critic, actor-critic, A2C, PPO, Q-learning, Q learning, Q value, Bellman, DQN, Deep Q-network, mujoco, OpenSim RL, PyGame Learning Environment, Unity ML, OpenAI Gym].

### A.2 ADDITIONAL LOGISTIC REGRESSION SUMMARIES

Table 5: **Return rate for first authors as a function of mean reviewer score** for men (top) and women (bottom). The logistic regression models have a 69.2% and 78.1% accuracy and a hold-out set containing 30% of 1,848 male and 30% of 243 female first authors. Mean reviewer score is a stronger factor on the return rate of women than men.

| Variable | Coefficient | Std. Error | Z-score | p-value |
|---|---|---|---|---|
| mean reviewer score | 0.1987 | 0.031 | 6.092 | 0.000 |
| constant | -1.5866 | 0.146 | -10.886 | 0.000 |
| mean reviewer score | 0.2356 | 0.097 | 2.43 | 0.015 |
| constant | -1.9602 | 0.437 | -4.488 | 0.000 |

Table 6: **Experience level of the first author on return rate** The logistic regression models have 69.5% and 67.84%, accuracy, respectively, on a hold-out set containing 30% of 2122 first authors.

| Variable | Coefficient | Std. Error | Z-score | p-value |
|---|---|---|---|---|
| first time author indicator | -1.1427 | 0.099 | -11.586 | 0.000 |
| mean reviewer score | 0.1820 | 0.030 | 6.120 | 0.000 |
| constant | -0.8021 | 0.153 | -5.241 | 0.000 |
| log base 10 publications | 0.1388 | 0.095 | 1.464 | 0.143 |
| mean reviewer score | 0.1927 | 0.030 | 6.451 | 0.000 |
| constant | -1.8050 | 0.183 | -9.859 | 0.000 |

Table 7: **Experience of the last author on the return rate of the first author** The logistic regression models have 64.78% accuracy on a hold-out set containing 30% of 2,122 first authors.

| Variable | Coefficient | Std. Error | Z-score | p-value |
|---|---|---|---|---|
| log base 10 citation | 0.1668 | 0.053 | 3.155 | 0.002 |
| mean reviewer score | 0.1788 | 0.035 | 4.888 | 0.000 |
| constant | -1.9845 | 0.223 | -8.891 | 0.000 |

Table 8: **Return rate for first authors as a function of a gender indicator.** The logistic regression model had a 70.8% accuracy and a hold-out set containing 30% of 2122 first authors.

| Variable | Coefficient | Std. Error | Z-score | p-value |
|---|---|---|---|---|
| gender indicator | 0.2104 | 0.152 | 1.381 | 0.167 |
| constant | -0.9977 | 0.144 | -6.928 | 0.000 |

Table 9: **Gender impact on acceptance**, ICLR 2021. Logistic regression predicting paper acceptance as a function of mean paper reviewer score and a gender indicator variable for both the first and last author. The model achieves 91% accuracy on a hold-out set containing 30% of 2021 papers. The results were inconclusive. Papers with first and last authors that were unlabelled were excluded.

| Variable | Coefficient | Std. Error | Z-score | p-value |
|---|---|---|---|---|
| mean reviewer score | 4.2505 | 0.182 | 23.312 | 0.000 |
| gender indicator, first author | -0.1363 | 0.215 | -0.635 | **0.525** |
| constant | -25.4785 | 1.098 | -23.210 | 0.000 |
| mean reviewer score | 4.2523 | 0.179 | 23.778 | 0.000 |
| gender indicator, last author | 0.0283 | 0.235 | 0.120 | **0.904** |
| constant | -25.6643 | 1.091 | -23.514 | 0.000 |

Table 10: **Country of origin impact on acceptance**, all submissions to ICLR from 2017 to 2021. Logistic regression predicting paper acceptance as a function of an indicator of what country it was from. The model achieves 70% accuracy on a hold-out set containing 30% of 2021 papers.

| Variable | Coefficient | Std. Error | Z-score | p-value |
|---|---|---|---|---|
| US | 0.2506 | 0.056 | 4.492 | 0.000 |
| Canada | 0.3011 | 0.120 | 2.518 | 0.012 |
| South America | 0.0770 | 0.602 | 0.128 | 0.898 |
| Australia/NZ | 0.0006 | 0.227 | 0.003 | 0.998 |
| Middle East | -0.2050 | 0.230 | -0.893 | 0.372 |
| UK/Ireland | -0.0233 | 0.147 | -0.159 | 0.874 |
| Mainland Europe | 0.0921 | 0.109 | 0.844 | 0.398 |
| Russia | 0.3259 | 0.579 | 0.562 | 0.574 |
| South Asia | -0.3468 | 0.132 | -2.632 | 0.008 |
| East Asia | -0.5442 | 0.119 | -4.578 | 0.000 |
| constant | -0.8879 | 0.035 | -25.347 | 0.000 |

Table 11: **Country of origin impact on acceptance**, all submissions to ICLR from 2017 to 2021. Logistic regression predicting paper acceptance as a function of mean reviewer score and an indicator of what country it was from. The model achieves 90% accuracy on a hold-out set containing 30% of 2021 papers.

| Variable | Coefficient | Std. Error | Z-score | p-value |
|---|---|---|---|---|
| mean reviewer score | 3.0962 | 0.075 | 41.426 | 0.000 |
| US | 0.0770 | 0.093 | 0.827 | 0.408 |
| Canada | 0.0714 | 0.196 | 0.364 | 0.716 |
| South America | 1.4728 | 0.973 | 1.514 | 0.130 |
| Australia/NZ | -0.0628 | 0.356 | -0.176 | 0.860 |
| Middle East | -0.1418 | 0.375 | -0.378 | 0.705 |
| UK/Ireland | 0.3342 | 0.241 | 1.384 | 0.166 |
| Mainland Europe | -0.1479 | 0.181 | -0.818 | 0.414 |
| Russia | -0.0601 | 0.833 | -0.072 | 0.942 |
| South Asia | 0.0807 | 0.236 | 0.341 | 0.733 |
| East Asia | -0.2593 | 0.185 | -1.401 | 0.161 |
| constant | -18.3354 | 0.440 | -41.656 | 0.000 |

Table 12: **Theorem impact on acceptance for each category**, all submissions to ICLR from 2017 to 2021. Logistic regression predicting paper acceptance for each category as a function of a theorem indicator. The models have a 68.72%, 74.83%, 66.91%, 68.63%, 70.11%, 72.12%, 66.66%, 69.68%, 63.63%, 70.19%, 70.96%, 61.4% accuracy on a hold-out set containing 30% of all papers.

Regressions are for topics in the following order: Theory, Vision, NLP, Adversarial, Generative Modelling, Meta-Learning, Fairness, Transformers, Optimization, Graph, Bayesian, and RL.

| Variable | Coefficient | Std. Error | Z-score | p-value |
|---|---|---|---|---|
| theorem indicator | 0.1612 | 0.097 | 1.667 | 0.095 |
| constant | -0.7764 | 0.071 | -10.904 | 0.000 |
| theorem indicator | 0.3625 | 0.273 | 1.327 | 0.185 |
| constant | -1.0557 | 0.110 | -9.580 | 0.000 |
| theorem indicator | 0.6300 | 0.205 | 3.066 | 0.002 |
| constant | -0.7925 | 0.077 | -10.271 | 0.000 |
| theorem indicator | 0.3160 | 0.129 | 2.454 | 0.014 |
| constant | -0.9762 | 0.075 | -12.969 | 0.000 |
| theorem indicator | 0.3597 | 0.142 | 2.536 | 0.011 |
| constant | -0.9391 | 0.074 | -12.616 | 0.000 |
| theorem indicator | 0.1627 | 0.257 | 0.633 | 0.526 |
| constant | -0.8372 | 0.100 | -8.334 | 0.000 |
| theorem indicator | 1.2009 | 0.489 | 2.456 | 0.014 |
| constant | -1.4523 | 0.335 | -4.336 | 0.000 |
| theorem indicator | 0.3199 | 0.197 | 1.624 | 0.104 |
| constant | -0.9705 | 0.084 | -11.511 | 0.000 |
| theorem indicator | 0.3474 | 0.406 | 0.856 | 0.392 |
| constant | -0.8755 | 0.376 | -2.326 | 0.020 |
| theorem indicator | 0.2986 | 0.164 | 1.821 | 0.069 |
| constant | -0.9793 | 0.091 | -10.758 | 0.000 |
| theorem indicator | 0.4187 | 0.271 | 1.547 | 0.122 |
| constant | -1.0622 | 0.155 | -6.852 | 0.000 |
| theorem indicator | 0.3792 | 0.221 | 1.715 | 0.086 |
| constant | -0.7056 | 0.137 | -5.163 | 0.000 |

Table 13: **Industry impact on acceptance, Google, Facebook, and Microsoft**, ICLR 2021. Logistic regression predicting paper acceptance as a function of mean reviewer score and 3 industry indicators. The model achieves 91% accuracy on a hold-out set containing 30% of 2021 papers.

| Variable | Coefficient | Std. Error | Z-score | p-value |
|---|---|---|---|---|
| mean reviewer score | 4.2521 | .106 | 40.006 | 0.000 |
| google | .5036 | .191 | 2.639 | .008 |
| facebook | -0.1219 | .291 | -.0419 | .675 |
| microsoft | .3420 | .269 | 1.270 | .204 |
| constant | -25.6433 | 0.636 | -40.350 | .000 |

