# OpenReview forum: "An Investigation into the Role of Author Demographics in ICLR Participation and Review"
_ICLR.cc/2022/Conference — ICLR 2022 Submitted_

### Official Review · Reviewer_b44g · 2021-10-30

**Correctness:** 3
**Technical Novelty And Significance:** 1
**Empirical Novelty And Significance:** 2
**Recommendation:** 5
**Confidence:** 4

**Main Review:**

The exploratory data analysis and regression results presented in the paper contain some interesting results. Although the authors do not provide explanations behind each of these results, it might be helpful to raise awareness of these patterns in the community. At the same time, most of the findings from this exploratory analysis overlap with those of (Tran et al., 2020), thus I am unsure of how novel the authors' contributions are, besides presenting an updated analysis.

As the authors correctly note in the Ethics Statement, this is an observational study. All regression models control only for the reviewers' scores, thus in many instances it's unclear what kind of effect, if any, these models are actually capturing. The paper contains statements such as "probe the effects of paper topic on the review process", "review score impacts retention", "how it impacts their return rate", "we remove the effect of reviewer score by controlling for this variable, there is still a small residual effect of gender" etc. These regressions simply uncover associational patterns and thus causal statements should be avoided.

About Section 3.1:
* In Section 3.1, the histogram on the left of Figure 1 would suggest a large difference in scores between men and women, so I found the fact that the difference was only 0.13 surprising. Is this difference of 0.13 based on the data whose distribution is displayed in the histogram?
* Was any residual diagnostics analysis for the logistic regression models conducted? All models are simple (i.e., they include only a few variables) so it should be rather straightforward to check whether the assumptions are met (e.g., by using the diagnostic plots presented here https://arxiv.org/pdf/1612.03257.pdf)
* In the regression of retention ~ reviewers scores+gender, how were observations of individuals that appeared in multiple papers as first authors handled? These observations are dependent and thus it might be a good idea to cluster the standard errors by the first author
* Are these results robust to changes in the way that reviewers scores are taken into account? e.g., if the most negative score instead of the mean score is accounted for, do the results still hold?
* Past work has shown that type I error may not be controlled for when running logistic regression on peer-reviewed data (https://arxiv.org/abs/1912.13188), so this issue should be acknowledged
* Based on Table 1, we cannot really say much about whether the difference between the coefficients in the two regression models is statistically significant. The most straightforward way to make some inference on that would be to fit a single model of retention ~ intercept + I(is woman) + reviewer score + reviewer score * I(is woman) and do a Wald test on the last coefficient. That said, the difference in the  probabilities predicted by the current models are very small
* some of the concerns and suggestions raised above apply also to the remaining sections of the paper


About Section 3.2:
* "we find that the number of publications of the first author had a positive impact on return rate (p = 0.143, Table 6)". This p-value is above any of the significance levels that are generally considered (0.001, 0.01, 0.05, 0.1), so I would interpret this coefficient (and not "impact") as not being statistical significant. I read that "borderline p-value 0.11", so I assume that the significance level the authors adopted is 0.1.

Minor details:
* some of the summary statistics that are reported contain two decimal digits, while others only contain one
* the manuscript contains some typos (e.g., "These data was")
* Section 1: In (Zweben & Bizot, 2021) I could not find the information about women comprising 23.6% of the enrollees in computation graduate programs
* Section 2: Perhaps it might useful to mention that the papers included in the dataset represent almost (but not) all of the submissions to ICLR in the period considered
* Captions of figures 3-7 need to be improved. Table 1 needs to be improved too, e.g., by adding an additional column on the left containing the labels "men" and "women"
* It would be helpful if the authors could provide the code to replicate these results

**Summary Of The Paper:**

The paper conducts an observational study of disparities in the ICLR 2017-2021 reviewing process. The work centers around the analysis of disparities in reviewing scores and acceptance rates across gender, number of authors, countries, papers topics, and industry/academia. The authors present several findings that are likely of interest to the community and provide some potential explanations behind some of such findings.

**Summary Of The Review:**

The analysis presented in the paper contains some interesting results. However, many of the findings overlap with those of (Tran et al., 2020), which conducted a similar analysis. In addition, I raised several concerns and questions on the methodology used in the paper.

---

> ### Author Response · Authors · 2021-11-22
> **Response to reviewer feedback**
>
> Avoiding claims that imply causal relationships:
>
> We agree that the statements you identified were ambiguous, and we did not intend to sound as if we have demonstrated causal relationships.  Thank you for pointing this out. We have made a careful pass on the paper to scrub these kinds of statements away.  We would point out that in some cases we use the word “effect” to refer to the statistical notion of effect size, which we do not think implies causation.
>
> Regarding first authors submitting multiple papers:
>
>  We did consider this issue, but we observed that over 80% of first authors had only one paper,  and so we decided to treat each paper like an independent sample so that we could use standard statistical methods that most readers would recognize.
>
> Alternative summary statistics for reviewer scores:
>
> In our own offline explorations, we did quite a bit of tinkering with using the median vs mean reviewer score but found it made little difference in our results for a range of studies.  To address your question specifically, we repeated the study of return rate as a function of reviewer score.  The original study found an effect size (coefficient of mean reviewer score) of 0.24 (p = 0.015).  If we instead conduct the regression using median or max reviewer score we get effects of size 0.179 and 0.14 (p<.05 for both).  Interestingly, we found that min score is a weak predictor or return rate, with an effect of -0.13.  This is likely because min score is not a reliable predictor or paper rejection, while max score is, i.e., a min score of 4 or 5 (weak reject) just makes a paper borderline, while having a max score of 5 almost guarantees rejection.
> Ultimately, we chose to use the mean because our experience is that the mean score is often used by ACs and SACs as a bar for calibrating acceptance rates, and our regressions behave similarly using either mean or median.
>
> False alarms:
>
> Thanks for making us aware of the work in https://arxiv.org/abs/1912.13188.  We think the “false alarm” issue is quite interesting, especially with respect to the issue of reviewer calibration.  We have added this to our discussion of limitations in the conclusion, and we added a citation in the intro.
>
>
> Combined version of table 1:
>
> You asked for a version of Table 1 that contains an indicator variable for gender.  Such as table was included in the appendix of the original submission.  We included two separate tables in the main paper because they showed that mean reviewer score has a statistically significant correlation with retention within both the women and men sub-populations.  The combined table you suggested shows that reviewer score is a significant factor for the population as a whole, however the population is about 90% men. It could be that this mean reviewer score is a statistically significant factor in the whole-sample regression, even if the variable has little or no effect for women (Note: it is actually possible to use various numbers from the combined table to calculate that the trend is significant within the population of women alone, although I wouldn’t expect a typical reader to do this).  We have modified the paper so that Table 1 contains the indicator variables you asked for, and we moved the analysis using two separate tables to the appendix.  Indeed, as you suspected, the difference in slope between men and women is not statistically significant (although women with rejected papers are significantly less likely to return to the conference, likely because their scores are lower - we comment on this in the new version of the paper).
>
> Use of the word “impact”:
>
> Section 3.2:  We agree that we should be more careful about when we claim an impact.   We would point out though that we included the p-value here deliberately to display that the evidence for this trend is weak.  We have revised this section of the paper and we no longer make any such claim.  We have also made a pass on the paper to remove claims of “impacts” or “effects” when the p value is not significant.

---

> > ### Comment · Reviewer_b44g · 2021-11-27
> > **Rebuttal response**
> >
> > I thank the authors for clarifying some of the points that I raised in my review. However,
> >
> > _Regarding first authors submitting multiple papers:_ If about 20% of the papers have common authors, then the dependence structure present in the data needs to be handled otherwise standard errors (such as those presented in this work) will be invalid. In my previous review, I have suggested using the same logistic regression model but clustering standard errors. This is a standard way to deal with the dependence.
> >
> > _Combined version of Table 1:_ According to the results of Table 1 in the revised version of the paper, the estimate of the coefficient relative to the interaction between gender and mean reviewer score is not statistically significant at the 0.1 significance level considered in the paper. However, there is no comment about this in the text. I am unsure about why Table 5 was kept in the paper now that Table 1 answers (in the most straightforward way) the question of interest. In addition, (i) in the tables it should be specified whether "gender" refers to woman or man and (ii) what does the "separate test" (p-value=0.073) in the same paragraph refer to?
> >
> > Lastly, as I have already mentioned, model diagnostics should be conducted. There is little reason to believe that a logistic regression accurately models the phenomena studied in the paper.

---

### Official Review · Reviewer_fQri · 2021-11-01

**Correctness:** 4
**Technical Novelty And Significance:** 2
**Empirical Novelty And Significance:** 3
**Recommendation:** 6
**Confidence:** 4

**Main Review:**

Strong points:

-The peer review process in machine learning, and diversity and/or bias issues in CS/AI/ML, are both crucially important issues that are in dire need of research.  This work addresses both of these issues simultaneously, and is therefore likely to obtain substantial attention.

-This work studies not only the impact of gender and geographic location, but also industry affiliation, the use of theorems, topics, experience level, etc.

-Substantial and labor-intensive manual annotation of ICLR authors' gender was performed in order to execute the study. The authors chose to perform this work to get more reliable results, instead of simply relying on automatic tools which are known to have a bias toward performing better on Western names.

-Attempting to control for various factors, due to the observational nature of the dataset, makes the analysis much more rigorous.

-There are several very interesting findings in the research, which could have real-world impact on the peer review process, and on the academy at large. For example: ICLR authors are more likely to return or have their paper accepted if their mentor is the same gender, larger ICLR teams have higher review scores on average, women tend to have larger co-author teams than men.

Weak points:

-The precise relationship to the work of Tran et al. (2020) is unclear, e.g. what are the methodological differences and the novel findings? (Note, if this paper turns out to be by the same authors as the Tran et al. arXiv paper, and is actually a conference version of that same paper, its significance would be higher. I cannot assess this due to the double blind.)

-It is a complicated question as to whether this paper is a good "fit" for ICLR.  On one hand, this work does not have the methodological novelty we would expect of an accepted ICLR paper, and this type of research is not normally published at this venue.  On the other, since it studies the ICLR conference itself, its findings are clearly of interest to the ICLR community.  Furthermore, it would behoove the ICLR community to be open to submissions such as this which are "outside the box" but still insightful.

-Arguably, a journal format would be more appropriate for this work, as it would provide more space for a thorough analysis. Then again, there is no journal which is directly affiliated with the ICLR conference, so this would have disconnected the paper from the ICLR community.

-I would like to have seen a more sophisticated level of analysis which leverages techniques developed by the ICLR community, and from machine learning at-large.  For example, NLP methods including topic models, word embeddings, and/or BERT could have contributed to the analysis.  This work only scratches the surface of the analyses on this dataset which are possible when leveraging the authors' gender annotations.

-The paper could be strengthened by a deeper investigation into whether the disparities identified are due to factors relating to systemic bias, sexism, racism, etc.  Similarly, the paper could have delved deeper into the extent to which English language capabilities are the reason for the disparities, e.g. by reporting results when controlling for the percentage of English language speakers per country.

-The potential impact of COVID-19 on the population under study should have been discussed.

-Since much of the analysis hinges on annotations, it would be appropriate to report inter-annotator agreement metrics.



Questions to the authors:

-Why is it particularly important to perform this study on the ICLR conference (other than the fact that the data are available on OpenReview)?



Additional feedback / minor suggestions:

-Pg 3, the phrase "limited evidence" is a bit ambiguous as it is unclear whether authors are saying there is evidence (the data suggest the statement is true, but the evidence is limited), or there isn't evidence (the data suggest the statement is false, but the evidence is limited)

-Pg 3, use parenthetical citations (\citep) for "Etzkowitz et al. (1992); Rosser & Lane (2002)"

-Pg 5, "increase monotonically with number of authors" (missing "the")

-Pg 6, "papers from East and South Asian" (Asia)

-Pg 7, "women did doing better than men" (remove "doing")

-Pg 9, "as a function 3 industry indicators" (of)

-Pg 9, "10.62%" (missing parentheses around the number)

-Pg 9, "mentorship is import" (important)

-The authors should mention that this work assumes a gender binary, and that future work could and should study nonbinary and transgender populations.

-Once this work is published, it would be extremely valuable if the authors could make their annotated dataset publicly available.

**Summary Of The Paper:**

This paper presents a study on the relationship between demographics and the submission and review of papers to ICLR, leveraging publicly available data from OpenReview.  The authors perform a substantial manual annotation effort on this dataset, and then study how paper acceptance and "retention" (i.e., whether authors submit to the conference again the following year) are related to gender and geographical location, by attempting to control for various factors. Topic, industry affiliation, experience level, and the use of theorems are also examined.

**Summary Of The Review:**

This paper does not fit the typical mold of an ICLR paper, in that it is mainly empirical work rather than presenting novel methods.  As always, more analysis could have been done.  However, its findings regarding participation and review in ICLR are valuable to the ICLR community, and it would likely obtain substantial attention in the literature and have an impact on the machine learning community at large.

---

> ### Author Response · Authors · 2021-11-22
> **Response to reviewer feedback**
>
> Thank you for your time and careful review.  We have fixed the typos and addressed the other edits you suggested.  We did consider using a number of NLP methods in this study, but after deliberation, we concluded that they did not make our results stronger.  For example, automated topic models create categories that are not necessarily interpretable, or do not have as well defined boundaries that correspond to the sub-fields of ML that most people think of, whereas hand-crafted rules gave us the ability to carve out interpretable fields like (e.g.) “reinforcement learning.”  We do however think that the interplay of writing style with acceptance rates is an interesting topic for future study.  We did consult with an expert on the issue of using language models to quantify writing proficiency (we thought this might help elucidate the differences between papers from different geographic regions), but we found that there was no existing framework that could give sensical or scientific measurements of writing “quality.”
>
> We added a statement to the section on gender labeling that makes clear the limitation of only considering binary labels.  We think the issues that non-binary and transgender people confront are important, but the small size of this sub-population at ICLR makes it difficult to do so using statistical methods.  The experiences of non-binary people in a community of this size are best studied at the level of individual experiences and stories, and not at the level of aggregated population statistics.

---

> > ### Comment · Reviewer_fQri · 2021-11-30
> > **Thank you for the response to the review**
> >
> > Thank you to the authors for the helpful response, which clarifies some things.  I appreciate the changes, which were positive as well. My score remains positive.

---

### Official Review · Reviewer_4yLt · 2021-11-02

**Correctness:** 3
**Technical Novelty And Significance:** 3
**Empirical Novelty And Significance:** 3
**Recommendation:** 6
**Confidence:** 4

**Main Review:**

STRENGTHS:
- This paper studies an important topic regarding how conference (here specifically ICLR) submission/score/acceptance is correlated with various aspects that are in and out of authors' control, including their choice of topic, gender, location, etc. This has implications for how conferences can be designed to be more inclusive, guide scholars toward more friendly/specific venues, and how as a community we may re-think the assignment of prestige/importance to conferences.
- The authors examine several cross-variable impacts, including gender/topic, experience/geographic region, and gender/experience. It would be interesting to measure how impactful such variables are compared to each other, perhaps via some sort of ANOVA or recursive elimination feature importance assessment, or the proposal of a method for measuring confounding/combinatorial impacts of each of the studied variables.

WEAKNESSES:
- It would have been interesting to see how gender representation could have changed if non-first/last authors were also labeled with their gender identity, although I recognize this may have been too expensive to label.
- It is very difficult in general to make strong comments about the impact of various demographic factors on conferences and the review process, and while the authors propose several important insights and hypotheses to explain differences in e.g. return rate of women, it is difficult to establish how likely these hypotheses are (or to establish causality)
- It would be interesting to see how the authors account for the self-selection bias of the fields/organizations that submit to or have members who attend ICLR, and perhaps compare this to other conferences (including those that are more specialized to specific fields like ACL, as well as other more general conferences like NeurIPS).
- There is not much discussion of possible reasons for the scoring difference among geographic regions; it would be nice to see more clarity around what causes could be hypothesized and what data the authors believe is missing/can be collected to test such hypotheses.

**Summary Of The Paper:**

This paper aims to quantify to some degree how gender, country of origin, and paper topic impact the review process, decision to submit, and paper review outcomes at ICLR, using data from 2017 to 2021. The authors report several interesting outcomes, including a lower return/retention rate for women attendees of ICLR (as well as authors with low review scores), higher scores for papers of Western origin, the increased likelihood of theorem-containing papers to be accepted, and the higher rate of acceptance for industry papers.

**Summary Of The Review:**

This paper touches on the very important topic of investigating explanatory effects of demographic and meta-factors in conference attendance, submission, and acceptance. This is an extraordinarily wide mandate, and the paper attempts to cover a fair amount of ground - this also results in several intuitive, unanswered questions when reading. Despite the opportunity for deeper analysis, I believe this paper is valuable in understanding representation at conferences (specifically ICLR) and provides some starting points for improving the same.

---

> ### Author Response · Authors · 2021-11-22
> **Response to reviewer feedback**
>
> Thank you for the constructive feedback.  We agree with some of the weaknesses you identified.  While we do think some of the factors we identified (such as retention rates and mentorship) likely do play a role in impacting demographics over time, it is difficult to establish causality, and we absolutely believe there are complex social factors at play that are outside the scope of our study.
>
> We have considered two possible reasons for the disparity in scores between papers from Asia and papers from the West.  First, we considered whether there was a way to measure the “quality” of written language in some meaningful way, but after consulting with an expert in this area, we concluded that there are no reliable scientific metrics for writing quality.  One factor we can measure is the level of experience of authors, which we did observe to correlate strongly with acceptance rates (i.e., we showed in the paper that first time authors have a lower acceptance rate than more experienced authors).  Interestingly, East Asia and South Asia have a much higher percentage of first-time authors (82% and 87%) compared to the US (66%),  and we think this experience gap likely contributes to the disparity in outcomes.  We have added a discussion of this to the paper.

---

### Official Review · Reviewer_kcQ1 · 2021-11-03

**Correctness:** 2
**Technical Novelty And Significance:** 1
**Empirical Novelty And Significance:** 1
**Recommendation:** 1
**Confidence:** 5

**Main Review:**

Positive aspects:
-	The observations are interesting, specially for the organizers and senior members of ICLR.
-	The methodology and analysis is clear and easy to follow.


Negative aspects:
-	The paper is not well-motivated and does not seem to fit within the general scope of ICLR (see more below).
-	The analyses are univariate and possibly misleading (see more below).
-	The study is non-conclusive and lacks a coherent structure.
-	No in-depth study of the observations is provided and the results raise more questions than answers.



Main Comments:

-	The main problem with the study is the focus on univariate analysis. The phenomena in social sciences often involve more than one variable and a univariate analysis gives a biased and often misleading answer. A notable example is the work by Blau and Kahn:
Blau, Francine D., and Lawrence M. Kahn. 2017. "The Gender Wage Gap: Extent, Trends, and Explanations." Journal of Economic Literature, 55 (3): 789-865.
For drawing meaningful conclusions, I encourage authors to perform a multivariate analysis on all the variables considered in this study and possibly more variables drawn from similar studies in social sciences.
-	The study is not well-motivate and it is not clear (1) how it fits withing the scope of ICLR, and (2) what conclusion should be drawn from the multiple correlations found in the study. The analysis done in this paper is narrowed to ICLR data only and does not relate to representation learning and broader applications. The paper is not conclusive and is built as a set of seemingly independent correlation-based analysis without a coherent structure.
-	There are no deeper analysis of the observations. For example, when a certain bias exists (assuming a correct analysis), what are the underlying dynamics that lead to these biases. Considering the anonymity of the reviewing process, in many cases where the authors found there is a gender bias in the accepted papers (per topic), the question arises that how can a gender “bias” appear in this kind of reviewing process. Isn’t this just a byproduct of the small dataset and even smaller subset of female authors within the dataset?
-	I find the justification behind considering only the first and last authors for the mentorship analysis rather poor. The papers submitted in CS venues are more often than not the product of more than one team’s effort in which the senior heads of each team do not necessarily have mentorship relationship with the first author of the paper (often times a student). I think authors should consider the affiliations, experience, and gender of all authors of a paper and draw conclusion on the mentorship. Also, there can be more than one mentor (e.g., two senior authors with the same affiliation as the first author). The same goes for attributing a location to a paper; last author is not a good representation necessarily. A better approach is to make a new class of “multi-national” papers.
-	In Section 3.1, authors study data from 2020 and 2021 to infer the statistics for first authors who came back the next year. The conference in these two years was virtual and, quite possibly, an anomaly in terms of the behavioral patterns. Authors should consider previous years as well.


**Summary Of The Paper:**

The is a data analysis study on ICLR submissions from 2017 to 2021. Several variables such as gender, location, prior experience, and topic of the papers is considered. The study is mainly univariate and tries to find meaningful patterns for inferring representation of minorities at ICLR.

**Summary Of The Review:**

I evaluate the paper as a reject for the following reasons:
-	The paper is not well-motivated and does not seem to fit within the general scope of ICLR (see more below).
-	The analyses are univariate and possibly misleading (see more below).
-	The study is non-conclusive and lacks a coherent structure.
-	No in-depth study of the observations is provided and the results raise more questions than answers.

---

> ### Author Response · Authors · 2021-11-22
> **Response to reviewer feedback**
>
> Relationship to “The Gender Wage Gap” by Blau and Kahn.
>
> Blau and Kahn’s landmark paper is able to do multi-factor studies because they use an economic survey with a sample size of 167,000.  Unfortunately, only 2,978 papers were submitted in 2021, and tests on a sample of this size have very low sensitivity when the data is further sliced along multiple axes at once.  Our paper contains both single factor and two factor studies that maintain a large enough sample size per category to find important trends, and to draw some conclusions with a high level of statistical confidence.
>
> “how it fits within the scope of ICLR”
>
> We feel that the broad audience which attends ICLR would be interested in our findings (as our work is about this very conference) and thus, we think ICLR is an appropriate venue.   We actually suspect that this paper is of interest to a larger percentage of conference attendees than an average accepted paper.  To quote reviewer fQri, “since it studies the ICLR conference itself, its findings are clearly of interest to the ICLR community. Furthermore, it would behoove the ICLR community to be open to submissions such as this which are "outside the box" but still insightful.”
>
> “The analysis done in this paper is narrowed to ICLR data only.”
>
> Some might interpret this as a feature and not a bug - we think it is reasonable to put a spotlight on issues at the conference level rather than aggregating several populations together. Still, we study ICLR because there is public access to all submitted papers and reviewer scores.
>
> “how can a gender “bias” appear in this kind of reviewing process. Isn’t this just a byproduct of the small dataset and even smaller subset of female authors within the dataset?”
>
> Given the statistically significant findings we present, it seems highly unlikely that these results arise simply as a random occurrence over a small sample size.  For example, women first authors with a woman last author have a higher retention rate than those with a male last author with a p-value of 0.02.  Men have a higher retention rate when paired with a male last author (p<0.001).
>
> Virtual conference in 2020 and 2021:
>
> This is a good point.  We have now added this to the discussion of limitations of our study.

---

> > ### Comment · Reviewer_kcQ1 · 2021-11-22
> > **Rebuttal Response**
> >
> > - **Multivariate analysis:** I don’t agree that the sample size of ~3k is not enough for multivariate analysis with decent precision. There are many methods proposed in this area for sample sizes not only smaller than 3k, but smaller than the dimension as well (e.g., https://www.sciencedirect.com/science/article/pii/S0047259X04000375). Another example is https://journals.plos.org/plosone/article?id=10.1371/journal.pone.0128379 , in which they show “Sample sizes as small as 58 individuals are sufficient for a broad range of multivariate abundance-based research”. As a final note and  as a rule of thumb, if the number of samples is 10 times the number of variables, it is possible to draw conclusions from multivariate analysis. I suggest authors look more closely to relevant literature on multivariate analysis to take full advantage of the valuable data they have collected.
> >
> > - **Relevance to ICLR:** Using ICLR as dataset does not automatically make a paper relevant to be published in ICLR. Although the results are interesting and can be of value for the senior members of the community, it does not overlap with ICLR scopes. It would help to make the study available privately or publicly to the ICLR area chairs to see, but certainly not as an ICLR-tier research paper in the conference.
> >
> > - **Unknown mechanism of gender bias found:** I am still not clear of what the authors try to propose regarding the claim of finding bias in the reviewing process. If the reviewing process is anonymous and yet there is a bias towards women, does it mean women write papers in a certain way that gives their identity (or at least gender) away? Or are they proposing that the process is not fully anonymized, and reviewers already know the identity of the authors?
> >
> > - **Generalizing observation from virtual ICLR to physical conference:** Considering the general conclusions drawn in the paper, the analysis such as that on virtual ICLR needs to be replicated for physical conference to have a conclusive observation. It is not a limitation that cannot be addressed in the paper and is directly related to the focus of the paper. I was expecting authors to replicate their experiments for physical ICLR data that they already have.

---

> > > ### Author Response · Authors · 2021-11-24
> > > **Our paper does not make claims of bias among reviewers**
> > >
> > >  Regarding the comment above about "the claim of finding bias in the reviewing process":
> > >    This is not an accurate characterization of our findings.  We do not claim to have found "bias" against women among reviewers.  Most of our work focuses on identifying factors that are correlated with retention rates for women. We do identify groups (including women) that, on average, receive scores below the conference mean. However these score discrepancies may be the result of sociological factors (e.g. differences in education, different peer group, etc...) and we do not make claims of any "bias" on behalf of reviewers.

---

> > > > ### Comment · Reviewer_kcQ1 · 2021-11-29
> > > > **Final Decision**
> > > >
> > > > The authors have not responded to my concerns in the "Rebuttal Response" post, which are crucial in justifying their methodology and reliability of their observations (lack of multivariate analysis and unjustified generalization using limited and possibly biased subset of the data). There are no connections made with the topics of interest in ICLR either. For these reasons, I am willing to keep my position to reject this paper.

---

### Decision · Program_Chairs · 2022-01-20

**Decision:**

Reject

**Comment:**

In this paper, the authors present an investigation of the impact of demographics on the peer review outcomes of ICLR. This is an important topic, as the demographics of ICLR and similar conferences are seriously skewed and may cause some people to feel excluded. The authors look into this complex problem with extensive manual annotations and analyses.

The main weakness of this paper is that it is observational, and while the results are interesting, it is difficult to take away a clear and convincing message for the future. Part of the reason is that the whole problem is quite complex, and the hypotheses that are presented and tested in this paper reveal relatively shallow findings. Compared to the NeurIPS experiments which are carefully designed, these are not causal (see one of the reviewers' comments), so it is difficult to draw conclusions beyond correlations.

In summary, the results are interesting, and despite some of the reviewers' concerns, I would not exclude this paper because of the topic being irrelevant to the cfp, but I think the paper needs a more clear and convincing message.